# Depletion of *Mdig* Changes Proteomic Profiling in Triple Negative Breast Cancer Cells

**DOI:** 10.3390/biomedicines10082021

**Published:** 2022-08-19

**Authors:** Chitra Thakur, Nicholas J. Carruthers, Qian Zhang, Liping Xu, Yao Fu, Zhuoyue Bi, Yiran Qiu, Wenxuan Zhang, Priya Wadgaonkar, Bandar Almutairy, Chunna Guo, Paul M. Stemmer, Fei Chen

**Affiliations:** 1Stony Brook Cancer Center, Renaissance School of Medicine, Stony Brook University, The State University of New York, Lauterbur Drive, Stony Brook, NY 11794, USA; 2Department of Pharmaceutical Sciences, Eugene Applebaum College of Pharmacy and Health Sciences, Wayne State University, 259 Mack Avenue, Detroit, MI 48201, USA; 3Department of Pathology, Renaissance School of Medicine, Stony Brook University, 101 Nicolls Road, Stony Brook, NY 11794, USA; 4Institute of Environmental Health Sciences, Wayne State University, 2309 Scott Hall, 540 E Canfield Ave, Detroit, MI 48202, USA; 5Department of Immunology and Microbiology, Wayne State University, Detroit, MI 48201, USA

**Keywords:** *mdig*, mass spectrometry, signaling pathways, breast cancer, biomarker

## Abstract

Triple-negative breast cancers are highly aggressive with an overall poor prognosis and limited therapeutic options. We had previously investigated the role of *mdig*, an oncogenic gene induced by some environmental risk factors, on the pathogenesis of breast cancer. However, a comprehensive analysis of the proteomic profile affected by *mdig* in triple-negative breast cancer has not been determined yet. Using label-free bottom-up quantitative proteomics, we compared wildtype control and *mdig* knockout MDA-MB-231 cells and identified the proteins and pathways that are significantly altered with *mdig* deletion. A total of 904 differentially expressed (*p* < 0.005) proteins were identified in the KO cells. Approximately 30 pathways and networks linked to the pathogenicity of breast cancer were either up- or downregulated, such as EIF2 signaling, the unfolded protein response, and isoleucine degradation I. Ingenuity Pathway Analysis established that the differentially expressed proteins have relevant biological actions in cell growth, motility, and malignancy. These data provide the first insight into protein expression patterns in breast cancer associated with a complete disruption of the *mdig* gene and yielded substantial information on the key proteins, biological processes, and pathways modulated by *mdig* that contribute to breast cancer tumorigenicity and invasiveness.

## 1. Introduction

Breast cancer is the second most common cause of cancer-related deaths in women in the U.S. after lung cancer. As of the year 2019, there are more than 3.1 million women with a history of breast cancer. An alarming rate as about one in eight women in the U.S. will develop invasive breast cancer during their lifetimes was noted in recent years [1]. Breast cancer is a clinically heterogeneous and highly complex disease composed of different biological subtypes such as the human epidermal growth factor receptor 2 (HER2), luminal A, luminal B, claudin-low, and basal-like [2]. The HER2, progesterone receptor (PR), estrogen receptor (ER), and the proliferation status as measured by Ki67 are the standard predictive and prognostic factors for breast cancers [3]. Among these subtypes, triple-negative breast cancer (TNBC) accounts for 10 to 20% of all breast cancer cases and is highly aggressive, and has the worst prognostic outcome. Lack of targeted therapy, metastatic spread, and relapse remain the top factors that make TNBC treatment challenging [4,5].

Several factors pertaining to genetics, epigenetics, environment, and lifestyle are involved in the etiology of breast cancer. These include mutations in the BRCA1 and BRCA2 genes, age, endogenous and exogenous exposure to hormones, obesity, alcohol consumption, and cigarette smoking as some of the known risk factors [6,7,8,9,10,11]. Understanding gene–environment interaction in breast cancer is a promising avenue of research. By studying the environmentally affected genes implicated in breast cancer, valuable information concerning the development and progression of breast cancers will be obtained. We have previously identified a gene named the mineral dust-induced gene (*mdig*, mina53, RIOX2), whose expression status influences the survival time of breast cancer patients. High expression of *mdig* predicted poor overall survival. However, for patients who are lymph node-positive, *mdig* expression is a favorable factor for prolonged overall survival [12]. Interestingly, suppression of *mdig* in breast cancer cells corresponds to enhanced methylation of DNA and histone protein, suggesting that the demethylase-like property of *mdig* is a factor in the pathophysiology. *Mdig* is likely to promote tumor growth in the early stages of cancer but acts as a tumor suppressor by inhibiting migration and invasion at the later stages [13]. After the initial discovery of *mdig* from the alveolar macrophages of coal miners exposed to mineral dust under occupational settings [14], several studies demonstrated increased expression of *mdig* in a variety of human cancers, especially cancers of the lung and breast [15]. *Mdig* also has a critical role in cell growth and motility [15], pulmonary inflammation [16,17], and immune regulation [18,19]. Cellular assays have shown a paradoxical role of *mdig* in cell proliferation, motility, and invasion in lung cancer [20], where *mdig*, being an environmentally induced gene, is induced upon exposures to certain environmental agents such as silica, arsenic, and tobacco smoke [21].

The development of TNBC and its related metastasis is a complex phenomenon that is poorly understood. Moreover, the role of *mdig* in aggressive breast cancers remains undefined. Very little is known about *mdig* except for its influence on breast cancer cell proliferation, migration, invasion, and DNA/histone methylation. Therefore, identifying key proteins modulated by *mdig* and the biological pathways operating in the development of breast cancers is pivotal. In the present study, we adopted a global proteomic approach to analyze the TNBC cells MDA-MB-231 that are knocked out for *mdig* via the CRISPR-Cas 9 gene editing technique. Wild type (WT) and knockout (KO) MDA-MB-231 clones were processed for high-resolution mass spectrometry, and the data were analyzed for the differentially expressed proteins. The underlying signaling pathways and prominent functions in cancer development were then evaluated. We have demonstrated significant pathways, protein networks, and the differential accumulation of critical proteins in the *mdig* KO cells, including Eukaryotic initiation factor 2 (EIF2) signaling, the unfolded protein response (UPR) signaling, upregulation of AKT, and ribosomal proteins. We also report some key proteins such as MAGED2, STMN1, HYOU1, PLAUR, RIN1, and SOD2 that have a role in predicting the overall survival in TNBC patients. Altogether, these results provide a strong basis for much-needed future research regarding *mdig*’s implication in breast cancers.

## 2. Materials and Methods

### 2.1. Cell Culture 

The human MDA-MB-231cells were purchased from the American Type Culture Collection (Manassas, VA, USA). MDA-MB-231 cells were cultured in DMEM F-12 medium. Cells were supplemented with 10% FBS and 1% penicillin–streptomycin (Sigma, St. Louis MO, USA) and grown in 37  °C-humidified incubators in the presence of 5% CO_2_.

### 2.2. Construction of the CRISPR-Cas9 Vector 

The procedures of CRISPR-Cas9 gene editing to knockout *mdig* were as reported previously [22]. Briefly, to generate the CRISPR-Cas9 plasmid, *mdig* CDS sequence was supplied into the CRISPR Design tool (http://crispr.mit.edu/, accessed on 17 July 2022), and single guide RNA (sgRNA) sequence targeting exon 3 of *mdig* was selected. The sense and antisense primer sequences are 5′-CACCGAATGTGTACATAACTCCCGC-3′ and 5′-AAACGCGGGAGTTATGTACACATTC-3′, respectively. Single-stranded sense and antisense primers were annealed to form double-strand oligos at 95 °C for 5 min, and then cooled down to 25 °C for 5 min. Vector pSpCas9-2A-Blast was digested with BpiI (BbsI) restriction enzymes (Thermo Fisher Scientific, Ann Arbor, MI, USA). sgRNA pairs and linearized vector were ligated by T4 DNA ligase (Thermo Fisher Scientific, Ann Arbor, MI, USA) for 10 min at 22 °C. Then the ligation product was transferred into DH5α competent *E. coli* strain (Thermo Fisher Scientific, Ann Arbor, MI, USA) according to the manufacturer’s protocol.

### 2.3. Transfection and Colonies Selection

MDA-MB-231 cells, 2.5 × 10^5^ /well in 6-well plate, were transfected with Lipofectamine 2000 (Thermo Fisher Scientific, Ann Arbor, MI, USA) according to the manufacturer’s protocol. Forty-eight hours after transfection, cells were sub-cultured in 10 cm dish for 24 h, followed by 2 μg/mL of Blasticidin (Thermo Fisher Scientific, Ann Arbor, MI, USA) selection for 2 weeks. Cell colonies were collected for screening of *mdig* expression by western blotting. Colonies without *mdig* knockout were used as WT cells, whereas colonies with successful *mdig* knockout were designated as KO cells.

### 2.4. Western Blotting

Total cellular proteins were prepared by lysing cells via sonication in 1 × RIPA buffer (Millipore, Billerica, MA, USA) supplemented with phosphatase/protease inhibitor cocktail and 1 mM PMSF. Lysed cells were then centrifuged and supernatant isolated as protein, which was quantified using the Micro BCA Protein Assay Reagent Kit (Thermo Scientific, Pittsburgh, PA, USA). Prior to loading onto SDS-PAGE gels, samples were boiled in 4 × NuPage LDS sample buffer (Invitrogen) containing 1 mM dithiothreitol (DTT). Samples were run on SDS-PAGE gels, and separated proteins were then transferred to methanol-wetted PVDF membranes (Invitrogen). Membranes were subsequently blocked in 5% nonfat milk in TBST and probed with the indicated primary antibodies at dilutions of 1:1000 overnight at 4  °C. The next day, membranes were washed with TBST and incubated with horseradish peroxidase (HRP)-conjugated secondary antibodies at dilutions of 1:2000 at room temperature for 1  h. Immunoreactive bands were visualized through SuperSignal™ West Pico Chemiluminescent Substrate detection system (Thermo Scientific, Rockford, IL, USA). *Mdig* (mouse) antibody was purchased from Invitrogen, and GAPDH was from Cell Signaling Technology (Danvers, MA, USA). The presented western blot data are representative of at least three independent experiments.

### 2.5. Preparation of Samples for Mass Spectrometry

Cell harvesting for proteomics analysis protocol has been adopted from [23]. Thereafter the samples in duplicates were submitted to the proteomics core facility of Wayne State University. In total, 34 cell pellets were submitted for proteomics analysis (that consisted of 5 WT and 12 KO samples in duplicates). Samples were weighed and volumes matched with the addition of HPLC-grade water. One percent LiDS final was added to the samples and heated at 95 °C for 5 min., followed by filtering through Pierce Handee Spin Columns (Thermo Fisher, Scientific, Ann Arbor, MI, USA) to remove non-soluble material. Protein amount was determined by BCA Protein Assay (range from 0.278 mg to 1.064 mg). A total of 50 µg aliquots of each were buffered with 100 mM ammonium bicarbonate (AMBIC), reduced with 5 mM DTT, and alkylated with 15 mM iodoacetamide (IAA) under standard conditions. Excess IAA was quenched with an additional 5 mM DTT. Overnight digestion was performed with sequencing-grade trypsin (Promega, Madison, WI, USA) in 100 mM AMBIC, 0.3 M urea, and 15% acetonitrile. The next day, detergent was removed from the samples using Pierce Detergent Removal Columns. Samples were speed vacced to dryness and solubilized in 0.1% formic acid for analysis. The peptides (4 µg-worth) were separated by reversed-phase chromatography (Easy Spray PepMap RSLC C18 50 cm column, Thermo Fisher Scientific, Ann Arbor, MI, USA), followed by ionization with the Easy Spray Ion Source (Thermo Fisher Scientific, Ann Arbor, MI, USA), and introduced into a Fusion mass spectrometer (Thermo Fisher Scientific, Ann Arbor, MI, USA). Abundant species were fragmented with collision-induced dissociation (CID).

### 2.6. Mass Spectrometry Data Analysis

For protein quantification and pathway analysis, mass spectrometry raw files were searched against the Uniprot human complete database downloaded 2017.07.14 using MaxQuant v1.6.2.10. Match between runs was enabled, and just one peptide was required for protein quantification. All other parameters were left at their default values. For all analyses, peptide spectra matches were accepted at a 1% false discovery rate as determined by a reversed database search.

### 2.7. Statistical Analysis 

Statistical analysis used R v3.4.3 (TIBCO, Palo Alto, CA, USA). Protein abundances were normalized to have the same median, and differential abundance between the wildtype and knockout samples was determined using a moderated *t*-test [24] with q-value correction for false discoveries [25]. In order to capture variability due to sample preparation and analysis, each sample was considered to be independent for statistical analysis.

### 2.8. Bioinformatics

Protein lists obtained from the MS data were processed using The Database for Annotation, Visualization, and Integrated Discovery (DAVID) version 2.0 (http://david.abcc.ncifcrf.gov/home.jsp, accessed on 17 July 2022). Functional enrichment analysis was carried out using the ClusterProfiler R package [26]. Proteins that were increased or decreased in KO vs. WT (*p* < 0.01) were submitted for analysis. QIAGEN’s Ingenuity Pathway Analysis (IPA^®^, QIAGEN, Redwood City, CA, USA, http://www.ingenuity.com/, accessed on 17 July 2022) software was used to investigate the functional and canonical pathways that were enriched in the differentially expressed proteins. Proteins that responded to *mdig* knockout (moderated t-test *p* < 0.005, *n* = 8) were submitted to IPA. All proteins identified in the study and pathways were considered significantly different with *p* < 0.05.

### 2.9. Kaplan–Meier Survival Analysis

A Kaplan–Meier survival database that contains survival information of breast cancer patients and gene expression data obtained by Affymetrix HG-U133 microarrays. The probe set for the indicated genes was used that scored to be the best among the other probe sets available by using JetSet best probe detection tool [27]. Survival curves resulting in *p* values of < 0.05 between the gene higher (gene^high^) and gene lower (gene^low^) groups were considered significantly different.

## 3. Results

### 3.1. Generation of Mdig Knockout Cells by CRISPR-Cas9 

To create *mdig* knockout cells, human TNBC cells MDA-MB-231 were transfected with a pSpCas9-2A-Blast vector containing sgRNA that targets the third exon of the *mdig* gene. Thereafter blasticidin selection was performed for two consecutive weeks, and the colonies obtained were screened for *mdig* expression by western blot (Figure 1A). Altogether we obtained 5 WT and 12 KO clones, and after screening them for *mdig* expression at the protein level, we prepared them for proteomic analysis. Each WT and KO clone was cultured and analyzed in duplicate. Two of the 34 samples were removed from further analysis for quality control reasons. A total of 5739 proteins were detected, and 5711 were quantified in at least one sample, among which 3569 proteins were quantified in all samples.

WT and KO MDA-MB-231 clones were submitted to high-resolution bottom-up mass spectrometry analysis using a Thermo Fusion mass spectrometer, and data were analyzed for the differentially expressed proteins. Eight samples from four WT clones and 18 samples from nine KO clones were submitted for differential expression analysis (*n* = 8). Clone KO#10 retained the expression of *mdig* and was removed from the analysis (Figure 1A). The principal component (Figure 1B) and cluster analysis (not shown) indicated some within-group heterogeneity. KO#3 and WT#5 were outliers from their groups. In order to reduce the potential impact of off-target effects, those samples were also removed from the analysis. Therefore, the clones KO#10, KO#3, and WT#5 were removed from the dataset and not used in any further analysis.

### 3.2. Identification of the Differentially Expressed Proteins for Their Gene Ontology Annotation 

LC-MS/MS data were analyzed to determine the fold change (FC) as a normalized ratio for KO compared to WT control cells. This first screening of the raw data identified a set of proteins for which abundances increase or decrease in the MDA-MB-231 KO cells. The analysis consisted of the unique protein IDs, with their fold change, *p* value, and t statistics as a function of KO/WT. Thereafter proteins that were increased or decreased in KO vs. WT (*p* < 0.01) were submitted for Functional Enrichment analysis. This analysis was based on a different Gene Ontology (GO) Consortium classification system for biological processes, molecular functions, and cellular components (Figure 2). The top 10 pathways are displayed in the figure.

In the Biological Processes, pathways associated with signal recognition particle (SRP) dependent co-translational protein machinery targeting membrane and endoplasmic reticulum and translational initiation were upregulated in the KO, whereas pathways associated with ribosome biogenesis, rRNA and ncRNA processing, and cellular respiration were downregulated in the KO cells. Newly synthesized proteins need to be delivered to their accurate destination, and hence their correct cellular compartment and localization are essential in maintaining cellular homeostasis. This can be brought by either during their biosynthesis, also referred to as “co-translational” or after their biosynthesis, also known as the “post translational” [28]. In this context, the co-translational translocation is the main pathway for the proteins to enter the endoplasmic reticulum, where a group of ribonucleoprotein complexes called the SRPs carry the co-translational delivery of proteins to their bona fide membrane confinement. The components of the SRP-dependent co-translational protein machinery have functional relevance in cancer, where certain SRP entities are upregulated. For example, in breast cancer, 7SL RNA has been observed in the extracellular vesicle (EV) mediated cell transfer. In fact, when the activated stromal EVs relay the 7SL RNA to the neighboring TNBC cells, antiviral signaling is induced that further lead to the increment in cancer growth and malignancy [29]. Moreover, a recent study on the single-cell transcriptomes of the TNBC datasets found SRP-dependent co-translational protein targeting membranes as the top-ranked module in their analyses [30], further strengthening the fact that the SRP-dependent co-translational protein machinery is integral to the TNBC phenotypes, as observed in our *mdig* KO TNBC cells.

Among the GO term for Molecular Functions, pathways associated with cell adhesion, cadherin binding, actin filament binding, and structural constituent of cytoskeleton were upregulated in the KO cells. However, catalytic activity acting on RNA, activities of helicase, RNA helicase, ligase, electron transfer activity, snoRNA, and coenzyme binding were downregulated in the KO category. In the Cellular Component GO term, cytosolic ribosome, focal adhesion, cell-substrate junction, ribosomal subunit, cytosolic large and small ribosomal subunit were upregulated in the KO cells, whereas mitochondrial matrix, preribosome, small subunit processome, mitochondrial inner membrane, mitochondrial nucleoid, mitochondrial protein complex was downregulated in the KO cells. 

These patterns of protein and pathway distribution suggest that *mdig* significantly affected the family of proteins that are essential for important biological and molecular processes such as cell adhesion, translation, mitochondrial and cytoskeleton machinery, and ribosomal biogenesis. The enrichment and upregulation of proteins and pathways in the KO cells, with functional relevance to cell adhesion, cadherin binding, and cytoskeleton, indicate that with *mdig* depletion, the cells attain a favorable phenotype that eventually facilitates their metastasis propensity. This further corroborates our previous study that revealed an increase in the migration and invasion of *mdig*-silenced breast cancer cells [13]. Additionally, this analysis also warrants a further investigation of the *mdig* perturbed pathways related to ribosome biogenesis, mitochondrial functions, and cellular respiration that may have important implications in breast cancer. 

### 3.3. Canonical Pathway Analysis Reveals Key Signaling Cascades Affected by Mdig 

We analyzed the protein abundance data from the differentially expressed proteins and identified the top ten proteins consisting of the greatest magnitude change in KO over WT MDA-MB-231 cells (Table 1). Once the differentially expressed proteins were identified, the next step was to query the major signaling pathways affected by *mdig* depletion in TNBC cells. Distinct alterations in the signaling pathways and networks comprise the pathogenic repertoire of the disease vs. healthy cells. Therefore, to identify the major biological pathways perturbed in breast cancer, we utilized the Ingenuity Pathway Analysis (IPA) Software (IPA; Ingenuity^®^ Systems, Qiagen) to interpret the differentially expressed proteins in terms of predominant canonical pathways and derivation of mechanistic networks. Canonical pathways are well-defined biochemical cascades resulting in unique functional biological consequences. Performing the canonical pathway analysis of our dataset via IPA revealed 501 canonical pathways. The top five canonical pathways (*p* < 0.05) according to the number of identified proteins (as shown in brackets) were EIF2 Signaling (54), Isoleucine Degradation I (8), UPR (12), Regulation of eIF4 and p70S6K Signaling (31), and Caveolar-mediated Endocytosis Signaling (14) (Figure 3A). Among the enriched pathways, EIF2 signaling was the topmost canonical pathway found in our analysis and has been elaborated, as shown in Figure 3B. Among this signaling cascade, eIF4E was found to be elevated. eIF4E is a key player in cap-dependent translation and whose overexpression results in the oncogenic transformation process. The regulation of mRNA translation has a critical role in cancer and tumor progression. mRNA translation is a complex process consisting of initiation, elongation, termination, and ribosome recycling, where the initiation stage represents the rate-limiting step of protein synthesis [31]. In this context, the eukaryotic translation initiation factors (EIFs) are the major regulators of the initiation stage, where they assist in the stabilization of the functional ribosomal complex near the start codon and provide the necessary regulatory machinery for the translation initiation. Among the EIFs, the eIF4E is essential for the cap-dependent translation initiation in eukaryotes [32]. In fact, the 5′ 7-methylguanosine (m^7^G) cap and the 3′ poly(A) tail are the two important features of the mRNA that guards the mRNA against degradation, thereby facilitating the initiation of the translation machinery. The establishment of the eIF4F complex on the 5′-mRNA cap structure is vital for the recruitment of mRNAs to ribosomes; it constitutes the cap-dependent translation initiation process [33]. Therefore, dysregulation of mRNA translation and eIFs have important implications in the pathogenesis of several cancers. The EIF2 complex serves as a principal axis of translation and engage in the various process of oncogenesis and malignant progression, where its overexpression is a common feature in many cancers [34]. Because eIF4E is the key regulator of translation initiation, it has crucial engagement in the oncogenesis process [35,36], and its aberrant expression has also been reported in many cancers [34]. In breast cancer, elevated expression of eIF4E protein was correlated with aggressive tumor phenotypes [37]; additionally, eIF4E has a significant role in driving the metastasis of breast cancer [38].

In TNBC patients, a high level of eIF4E is associated with an unfavorable prognostic outcome [39]. Interestingly, PI3K and AKT were also upregulated with *mdig* depletion. Previous reports identified the PI3K-Akt pathway as an enhancer of the expression of epithelial and mesenchymal transition (EMT) resultant transcription factors such as Snail, Slug, ZEB1, and ZEB2 that promoted the EMT and resulted in an elevation of the cancer cell motility [40,41]. This suggests an increased motility potential of breast cancer cells upon the loss of *mdig* protein. 

### 3.4. Upregulation of eIF4-p70S6K and UPR Pathways in Mdig KO Cells

The above data suggested upregulation of EIF2 signaling in the *mdig* KO cells. Since eIF4 and p70S6K are important components of EIF2 signaling, we further elaborated on these two networks in the following analysis (Figure 4A). Analyzing the protein profiles belonging to the canonical pathway of eIF4 and p70S6K signaling revealed an increase in eIF4E, eIF3, and ribosomal protein RPS6 in the KO cells. In contrast, eIF4A, an mRNA helicase and a vital component of the eukaryotic translation initiation eIF4F complex, and integrin, was found to be downregulated. It is well known that activation of translation initiation is indispensable for the malignant phenotype, and dysregulation of protein synthesis contributes to the process of tumorigenesis. Meanwhile, integrin proteins are important for extracellular matrix remodeling and metastasis processes and are often downregulated in breast cancer [42]. Moreover, ribosomal proteins are RNA binding proteins that are primarily implicated in the regulation of protein translation. Alterations in ribosomal biogenesis and function have been associated with the acquisition of an aggressive tumor phenotype in breast cancer [43]. In this context, RPS6 is used as a readout of the mTORC1 activity in many diseases, including cancer, and increased phosphorylation and/or overexpression of RPS6 has been reported in breast cancer [44]. Interestingly, in mammalian cells, the PI3K/AKT/mTORC1/S6K pathway is the major signaling pathway that upregulates the phosphorylation of RPS6 [45].

The UPR signaling machinery has been widely studied in human carcinogenesis and tumorigenesis, where it regulates several processes predominantly associated with the classical “hallmarks of cancer” [46]. IPA pathway analysis indicated downregulation of the UPR signaling in *mdig* KO cells, except for the tumor necrosis factor-receptor-associated factor 2 (TRAF2) that was upregulated in the KO cells (Figure 4B). In breast cancer, TRAF2 has been shown to be able to promote breast tumorigenesis [47], as well as its increased expression correlated with invasion and metastasis [48,49]. Other proteins implicated in breast cancer, such as the Programmed cell death 1 (PD1) and GRP 94, were downregulated in the KO cells. The individual differentially regulated proteins in the top five canonical pathways, thus, certainly are potential targets for further investigation where *mdig* can be directly involved in ribosome biogenesis, UPR signaling, and the metastasis of triple-negative breast cancers.

### 3.5. Disease Pathways Regulated by Mdig

Disease-based pathway analysis using IPA analytic tool also revealed top enriched proteins associated with diseases and disorders in the categories of Cancer, Organismal Injury and Abnormalities, Tumor Morphology, Cardiovascular Disease, and Developmental Disorders in the *mdig* KO MDA-MB-231 cells. Interestingly, the top network identified was associated with cancer. This network consists of 458 proteins in our proteomic dataset (Figure 5A). These results indicate the strong involvement of *mdig* in regulating the process of malignant transformation of the breast cells. With Cancer being the topmost populated entity, as revealed in this analysis, *mdig* can be undoubtedly linked to the carcinogenic processes of breast cancer, such as cell motility, invasion, and metastasis, further corroborating our previous studies [13].

Furthermore, to evaluate the interaction networks that were affected by *mdig* KO, we also identified 25 networks built with 35 focus molecules in each network from the IPA’s knowledge base. The five most affected gene networks and a detailed interaction in the most significant networks have been shown in Figure 5B,C. Genes with different expression patterns predominantly mapped to the networks associated with protein synthesis, RNA post-transcriptional modification, DNA replication, recombination, and repair, which indicated an important role of *mdig* in regulating the genes associated with genomic stability and cancer.

### 3.6. Prognostic Implication of the Top Mdig-Regulated Genes in Breast Cancer

After having established the changes in protein abundance in *mdig* KO cells, we selected the top upregulated and downregulated proteins following *mdig* depletion for their clinical implications in human breast cancer. The rationale for selecting these proteins was based on the top-ranked pathways and their specific relevance in the pathogenesis of breast cancer, as indicated in previous reports. The most upregulated proteins in *mdig* KO cells consisted of MAGED2 and STMN1, while the most downregulated proteins include PLAUR, HYOU1, SOD2, and RIN1. The overall survival (OS) analysis for 618 TNBC patients in the Kaplan–Meier database [50] suggests that high expression of MAGED2 and STMN1 predicted poor OS, whereas increased expression of HYOU1, PLAUR, RIN1, and SOD2 predicted better OS of these patients (Figure 6). The differential regulation of such proteins by *mdig* is an important finding, where the involvement of these proteins in the regulation of cell proliferation, motility, invasiveness, cancer metabolism, and endoplasmic reticulum stress make them ideal candidates that can be exploited for therapeutic targeting. This also suggests that these proteins are the potential candidate biomarkers that are strongly associated with the tumorigenesis of TNBC or other types of aggressive breast cancers.

## 4. Discussion

Triple-negative breast cancer is a malignant form of breast cancer with aggressive clinical characteristics. It has the worst patient prognosis and currently lacks targeted therapy [51,52]. Therefore, there is an urgent need to understand the molecular and biological mechanisms governing the malignant behavior of TNBC and its pathogenicity. Initially, *mdig* was identified as an oncogene for lung cancer [53], and it was also expressed in other cancer types with roles in cell growth and motility [15]. In breast cancer, we found that high *mdig* expression predicts poor overall survival of patients. However, a high level of *mdig* predicts a better survival for the patients who had lymph node or distal organ metastasis, further suggesting that *mdig* may be favorable for metastatic patients [12]. In TNBC cells MDA-MB-231, silencing *mdig* by siRNA approach revealed important attributes of *mdig* where it reduces the DNA and histone methylation and the migration of the cells [13]. These studies indicated that *mdig* is important for the tumor growth of early-stage breast cancers, but at the later advanced stage, *mdig* expression is likely to benefit as it inhibits the migration and invasion of breast cancer cells. 

Our previous studies on TNBC cells silenced for *mdig* via short interfering RNAs (siRNAs) have yielded some important regulatory effects of *mdig* on cell motility, invasion, and DNA and histone methylation [13]. Though that model is a transient knockdown for *mdig*, the data generated suggest that *mdig* negatively regulates breast cancer cell migration and invasion potential. Still, the mechanisms underlying the influence of *mdig* on breast cancer cells are poorly understood and have not previously been explored, at least at the system levels with proteomics technology.

To gain a better understanding of the function of *mdig* in cell growth, motility, and invasion in breast cancer, in the current report, we applied the CRISPR-Cas9 gene editing technique to knockout *mdig* in human TNBC MDA-MB-231 cells, followed by a global proteomic analysis of the WT and *mdig* KO cells. Analyses of these data revealed some significant findings related to differentially expressed proteins and signaling pathways affected by *mdig*. Loss of *mdig* resulted in an increase in the abundance of proteins that are implicated in cell proliferation, angiogenesis, and metastasis of breast cancer. Among them are Cathepsin D, MAGED2, Filamin A, STMN1, and RACK1. The cathepsin family of proteins is known to mediate the metastasis of cancer cells and degrade the extracellular matrix and collagen. Cathepsin D is also overexpressed in breast cancer [54,55,56] and predicts a poor prognosis [57,58,59]. It represents a marker for invasive potential and aggressive behavior in high-grade carcinomas [60] and stimulates cell growth, angiogenesis, and metastasis [61,62,63]. MAGED2 is found to be elevated in primary tumors and is upregulated in metastasis [64]. It is noteworthy that in the present report, we identified MAGED2 as a novel protein that is increased in response to *mdig* knockdown. STMN1 is a microtubule destabilizing protein whose expression is associated with breast cancer proliferation [65,66]. In breast cancer patients, high STMN1 correlates with poor prognosis [67,68]. Moreover, elevated STMN1 is linked to high histological grade and low ER, PR expression status [65], and the aggressive phenotypes accompanied with cancer stem cell marker expression of breast cancer [69]. Interestingly, another protein that is associated with cell growth, adhesion, invasion, and metastasis is the RACK1 which was upregulated in response to *mdig* knockout. In fact, both in vitro and in vivo studies have shown that RACK1 promotes the proliferation, invasion, and metastasis of breast cancer [70], and it remains one of the independent predictors of poor clinical outcomes in breast cancer [71]. In fact, RACK1 was not only associated with breast cancer malignancy, but its overexpression is implicated in the growth and metastasis of several other cancer types such as lung cancer, gliomas, colon cancer, prostate cancer, liver cancer, epithelial ovarian cancer, and squamous cell carcinoma of the esophagus [72]. Finally, the top five proteins found to be upregulated in our analysis included Filamin A as well. Several studies reported the overexpression of Filamin A to be associated with highly metastatic cancers of the prostate [73], skin [74], and brain [75], and that Filamin A is involved in the progression of neoplasia [76].

Carcinogenesis is a multistep process, and alterations in signaling networks resulting from genetic, epigenetic, and environmental changes are accumulated at distinct stages of cancer development. In fact, dysregulated mRNA translation is critical in influencing the etiology and pathogenesis of human malignancies. Hence it is widely reported that aberrant translation of oncogenes, tumor suppressors, and eukaryotic translation initiation factors are paramount to the proliferation of cancer cells [77]. Our analysis showed the top signaling pathways that were found to be enriched in the TNBC cells upon *mdig* knockout is eIF4E, the key component of the EIF2 signaling. Studies reported that suppressing eIF4E significantly reduced the migratory and invasive potential of breast cancer cells and metastasis of the breast cancer cells in a mouse model [38]. In TNBC patients, a high level of eIF4E is associated with an unfavorable prognostic outcome [39]. We also found an elevated level of PI3K and AKT in the KO cells under this pathway analysis. It has been known that the PI3K-Akt pathway facilitates the expression of EMT-related transcription factors Snail, Slug, ZEB1, and ZEB2, thereby promoting EMT and enhancing the motility of cancer cells [40,41]. 

Among the proteins that were downregulated upon *mdig* knockout were the family of proteins implicated in tumor suppressor functions and stress-related cellular response. These downregulated proteins were PLAUR, SOD2, RIN 1, HYOU1, and NAMPT. PLAUR expression has been found in aggressive breast cancers such as in TNBC, a subset of Her2 positive breast cancer, and in tamoxifen refractory breast cancer [78,79,80]. It is intriguing that we also observed decreased expression of manganese superoxide dismutase 2 (SOD2) in the KO cells. Loss of SOD2 represents a phenotype of tumor initiation and, therefore, is indicative of the tumor suppressor role of SOD2, particularly due to its scavenging role for superoxide anion during the process of tumorigenesis [81]. Apparently, decreased SOD2 activity and hiked-up reactive oxygen species are the prerequisite for the metabolic reprogramming of cancer cells [82]. Additionally, forced SOD2 overexpression in cancer cells is able to decrease the metastatic potential [83,84]. In breast cancer, SOD2 is epigenetically regulated, where SOD2 expression is repressed primarily due to the hypo acetylation of histone proteins [85]. Moreover, there is a switch from SOD2 to SOD1 during the transformation process in breast cancers [86], and SOD2 is downregulated in malignant breast cancer cells compared to their normal cell counterparts [87]. 

The heterogeneity of the TNBC and lack of effective therapeutic targets, along with insufficient predictive biomarkers, are the underlying reasons for the challenges associated with TNBC therapy. The high throughput proteomics study of the TNBC cells in the present report has allowed us to stratify systemic differences between the MDA-MB-231 cells with and without *mdig*. Hence these proteins may serve as additional biomarkers in the prognosis of the TNBCs. Further studies on the mechanistic regulation of these proteins by *mdig* are certainly well-warranted. Taken together, the data presented here have provided a bioinformatical insight into the TNBC in context to *mdig* deletion, which has laid the foundation for identifying additional biomarkers and their functional implications for a better understanding of the development of aggressive breast cancers.

## Figures and Tables

**Figure 1 biomedicines-10-02021-f001:**
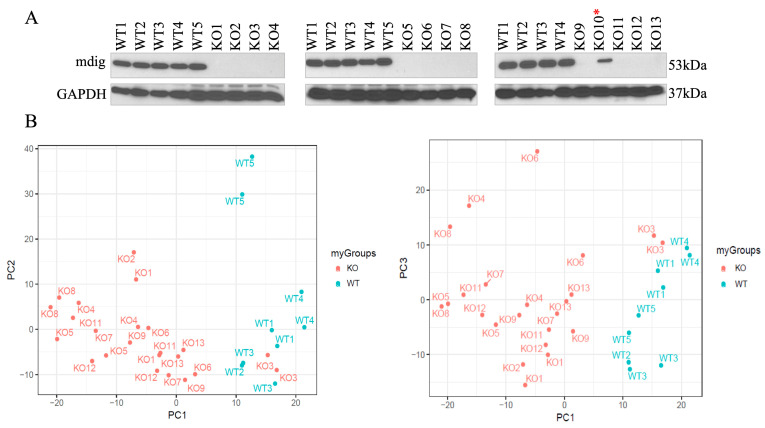
Establishment of *mdig* knockout cells by CRISPR-Cas9 gene editing. (**A**), Validation of *mdig* depletion by CRISPR-Cas9 gene editing through Western blotting for *mdig*. WT cells represent cells subjected to gene editing but *mdig* gene remains intact, whereas KO cells represent successful knockout of *mdig* by gene editing. A total of 5 WT and 13 KO clones are shown. GAPDH was used as a loading control. Asterisks (*) indicates the KO clone containing residual *mdig* protein. (**B**), All the WT and KO colonies in duplicates were subjected to proteomics assay. Principal component analysis of the colonies indicated some within group heterogeneity. The colonies were validated and were proceeded for downstream data analysis. KO#10, KO#3, and WT#5 were excluded from further analysis.

**Figure 2 biomedicines-10-02021-f002:**
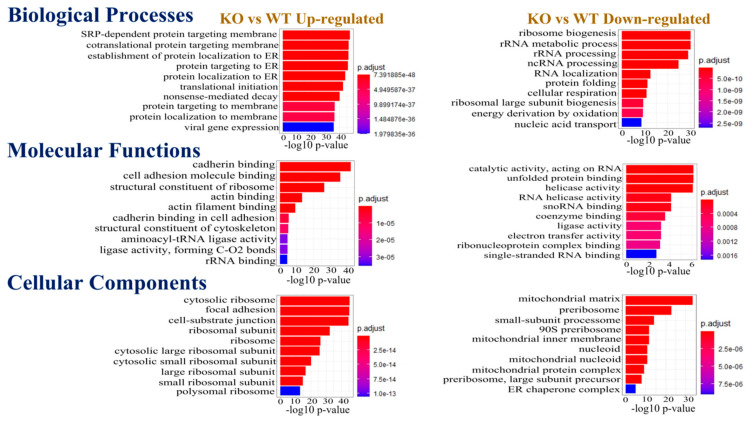
Functional enrichment analysis of differentially expressed proteins between WT and KO MDA-MB-231 cells. Gene Ontology Consortium for the Biological Processes, Molecular Functions, and Cellular Components, respectively, were used in this analysis. Data show the top 10 pathways. ER: endoplasmic reticulum, C: carbon, O_2_: oxygen.

**Figure 3 biomedicines-10-02021-f003:**
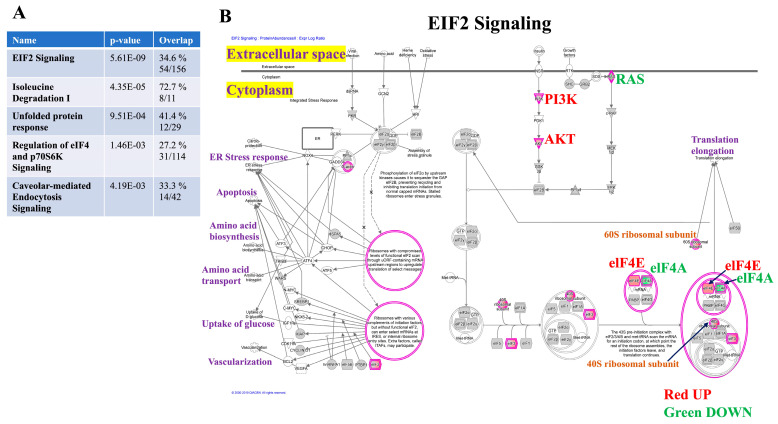
Canonical pathways identified by Ingenuity Pathway Analysis (**A**), Illustrated are the top five canonical pathways (*p* < 0.05) based on 904 differentially expressed proteins in the *mdig* KO vs. WT MDA-MB-231 cells. (**B**), Canonical pathway analysis revealed the eukaryotic initiation factor 2 signaling pathway among the topmost canonical pathways enriched upon *mdig* KO. The EIF2 Signaling has been elaborated by showing the upregulated and downregulated proteins and their cellular localization. Nodes represent molecules in a pathway, whereas the biological relationship between nodes is represented by a line (edge). Edges are supported by at least one reference in the Ingenuity Knowledge Base. The intensity of color in a node indicates the degree of up (red) or down (green) regulation.

**Figure 4 biomedicines-10-02021-f004:**
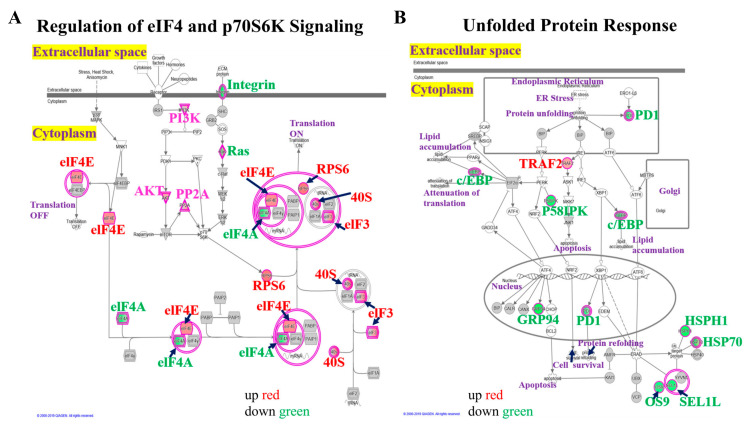
Enrichment of eIF4/p70S6K and unfolded protein response pathways in the KO cells. Canonical pathway analysis revealed the (**A**), regulation of eIF4 and p70S6K signaling pathway and (**B**), the UPR pathway among the topmost canonical pathways enriched upon *mdig* knockout. Nodes represent molecules in a pathway, whereas the biological relationship between nodes is represented by a line (edge). Edges are supported by at least one reference in the Ingenuity Knowledge Base. The intensity of color in a node indicates the degree of up (red) or down (green) regulation.

**Figure 5 biomedicines-10-02021-f005:**
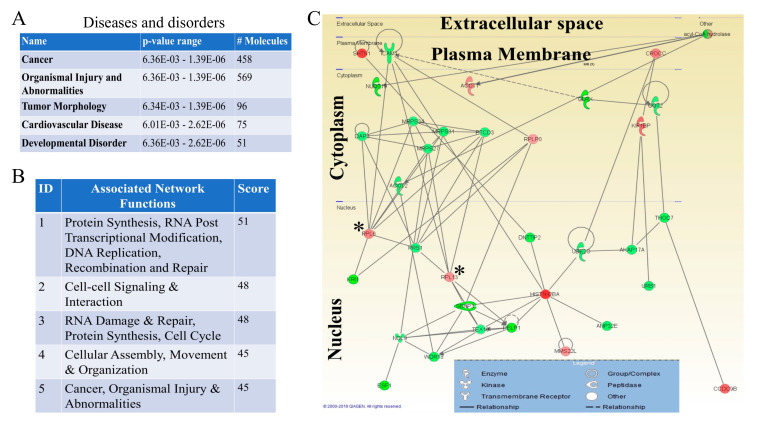
Diseases-associated pathways and protein network in the *mdig* KO vs. WT MDA-MB-231 cells as analyzed by IPA. (**A**), Top Diseases and Disorders, depicts the affected functional categories based on differentially expressed proteins where the Name column displays a category of diseases and functions showing Cancer, Organismal Injury and Abnormalities, Tumor Morphology, Cardiovascular Disease and Developmental Disorder as the top diseases in the proteomics dataset, along with the number of molecules associated with the individual diseases and disorder. (**B**), IPA of protein networks in the MDA-MB-231 KO over WT cells. The top five significant networks as determined by the IPA with their scores and associated functions has been shown. (**C**), Protein network showing the relationships and connectivity between the focus molecules of the network related to protein synthesis, RNA post transcriptional modification, DNA replication, Recombination and Repair. Red indicated increased and green indicates decreased expression. Star * represents the elevated ribosomal proteins in this pathway.

**Figure 6 biomedicines-10-02021-f006:**
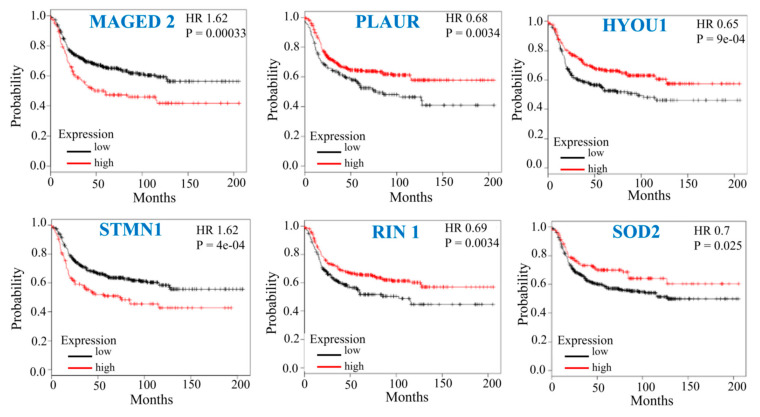
Prognostic implication of the selected top differential expressed protein in *mdig* KO vs. WT MDA-MB-231 cells. Top differentially expressed proteins (*p* < 0.05, *n* = 8, moderated *t*-test) as revealed by the proteomics study were evaluated for their prognostic significance in breast cancer. Survival curves were plotted for TNBC patients (*n* = 618) for the indicated proteins using Kaplan–Meier Plotter. X axis shows time in months and Y axis shows the overall survival probabilities.

**Table 1 biomedicines-10-02021-t001:** Top 10 most differentially expressed proteins in *mdig* KO vs. WT MDA-MB-231 cells (*p* < 0.005), as determined by the proteomics dataset obtained through mass spectrometry.

Symbol	Protein ID	Description	Fold Change
CTSD	P07339	Cathepsin D	11.063
MAGED2	Q9UNF1	Melanoma-associated antigen D2	10.443
FLNA	P21333	Filamin-A	9.662
ABHD16A	O95870	Abhydrolase domain-containing protein 16A	9.595
STMN1	P16949	Stathmin1	9.118
NPC2	P61916	Epididymal secretory protein E1	8.845
RACK1	P63244	Receptor of activated protein C kinase 1	8.820
HIST1H2BA	Q96A08	Histone H2B type 1-A	8.673
IQGAP1	P46940	Ras GTPase-activating-like protein IQGAP1	8.603
HUWE1	Q7Z6Z7	E3 ubiquitin-protein ligase HUWE1	8.473
*Mdig* (RIOX2)	Q8IUF8	Mineral dust-induced gene	−8.521
KRI1	Q8N9T8	Protein KRI1 homolog	−8.272
CCDC51	Q96ER9	Coiled-coil domain-containing protein 51	−8.241
PLAUR	Q03405	Urokinase plasminogen activator surface receptor	−8.203
HYOU1	Q9Y4L1	Hypoxia upregulated protein 1	−7.368
SOD2	P04179	Superoxide dismutase [Mn], mitochondrial	−7.173
RIN1	Q13671	Ras and Rab interactor 1	−7.096
NOP58	Q9Y2X3	Nucleolar protein 58	−7.083
NAMPT	P43490	Nicotinamide phosphoribosyltransferase	−7.006
MCCC2	Q9HCC0	Methylcrotonoyl-CoA carboxylase beta chain, mitochondrial	−6.863

## Data Availability

The datasets generated for this study can be found in the ProteomeXchange Consortium via the PRIDE [88] partner repository with the dataset identifier PXD016688 and 10.6019/PXD016688. Western blots have been deposited to the Biostudies database (https://www.ebi.ac.uk/biostudies/, accessed on 1 December 2021) with the accession number S-BSST333.

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
