# Peer review of "Depletion of *Mdig* Changes Proteomic Profiling in Triple Negative Breast Cancer Cells"

_biomedicines, 2022, doi:10.3390/biomedicines10082021_

Round 1
Reviewer 1 Report
The manuscript entitled "Depletion of mdig changes proteomic profiling in triple-negative breast cancer cells " by Thakur et al. presented a study on protein expressions in breast cancer with depletion of mdig gene. In addition, further information on the proteins, biological processes, and pathways regulated by mdig that lead to breast cancer oncogenesis and metastasis is also presented. This manuscript is clearly written, a detailed description of the methods, results and discussion make this manuscript easy to follow. I would like to recommend its publication after addressing the following comments:
- Please improve the resolution of Figures 3 and 4 to have a better quality to be reviewed.
- Please carefully check the appearance of abbreviations and full names of the proteins: some of the abbreviations have been shown, but the full names are still presented.
- Please mark the molecular weight on the western blot in figure 1A while the supplementary figure has provided.
Author Response
Reviewer 1
The manuscript entitled "Depletion of mdig changes proteomic profiling in triple-negative breast cancer cells " by Thakur et al. presented a study on protein expressions in breast cancer with depletion of mdig gene. In addition, further information on the proteins, biological processes, and pathways regulated by mdig that lead to breast cancer oncogenesis and metastasis is also presented. This manuscript is clearly written, a detailed description of the methods, results and discussion make this manuscript easy to follow. I would like to recommend its publication after addressing the following comments:
- Response: Dear reviewer, thank you for your positive feedback on our manuscript. We appreciate your comments and have complied to all your suggestions.
Please improve the resolution of Figures 3 and 4 to have a better quality to be reviewed.
- Response: We have provided the original high-resolution image that we downloaded from the IPA analysis. The improved figures 3 and 4 [ i.e., 3A & 3B and 4A & 4B separately] have been provided. Thank you.
Please carefully check the appearance of abbreviations and full names of the proteins: some of the abbreviations have been shown, but the full names are still presented.
- Response: We have carefully checked for the abbreviations and have corrected those whose full names were still present. The corrections can be viewed with the track changes highlighted in our revised manuscript. Thank You.
Please mark the molecular weight on the western blot in figure1A while the supplementary figure has provided.
- Response: We have indicated the molecular weight of 53kDa for mdig and 37kDa for GAPDH in the revised figure 1A. Thank you.
Reviewer 2 Report
In the paper by Thakur et al., the authors using quantitative proteomics, compared wildtype control (WT) and mdig knockout (KO) triple negative breast cancer (TNBC) MDA-MB-231 cells and identified the proteins and pathways that are significantly altered with mdig deletion. The role of mdig had previously investigated, as an oncogenic gene involved in the pathogenesis of breast cancer. They found that loss of mdig resulted in an increase in abundance of proteins that are implicated in cell proliferation, angiogenesis, and metastasis of breast cancer. Among them are Cathepsin D, MAGED2, Filamin A, Stathmin1 (STMN1) and RACK1. Among the proteins that were downregulated upon mdig knockout were the family of proteins implicated in tumor suppressor functions and stress related cellular response. These downregulated proteins were PLAUR, SOD2, RIN 1, HYOU1, and NAMPT. The proteomics study of the TNBC cells in this article allowed the authors to stratify systemic differences between the MDA-MB-231 cells with and without mdig. They concluded by stating that these proteins may serve as additional biomarkers in the prognosis of the TNBCs
Only few MINOR CONCERNS should be addressed by the Authors, before a publication on Biomedicines can be granted:
· In the paragraph 3.1: the authors should explain why they removed WT#5 from the dataset and they did not use it in any further analysis;
· In the paragraph 3.2: the authors should briefly describe the pathways associated with SRP dependent co-translational protein machinery that they cyte in the text;
· In the paragraph 3.3: the authors should briefly describe the cap-dependent translation and what its upregulation causes in the oncogenic transformation process;
· Fig. 3b and 4: the images have to be enlarged in order to see proteins’ names or their names in green/red need to be smaller because it is difficult to understand the pathway. Moreover, the purple can be confused with red, please change colour, and the intensity of colours are not clear. The authors have to improve these figures;
· Check typing errors;
· Always explain the acronyms the first time the authors cited them throughout the text.
Round 2
Reviewer 1 Report
Thanks for answering/addressing the comments. I would like to recommend its publication after careful proofreading.